# Effect of Quercetin on mitoBK_Ca_ Channel and Mitochondrial Function in Human Bronchial Epithelial Cells Exposed to Particulate Matter

**DOI:** 10.3390/ijms24010638

**Published:** 2022-12-30

**Authors:** Adrianna Dabrowska, Miroslaw Zajac, Piotr Bednarczyk, Agnieszka Lukasiak

**Affiliations:** Department of Physics and Biophysics, Institute of Biology, Warsaw University of Life Sciences, 02-776 Warsaw, Poland

**Keywords:** epithelium, mitochondria, mitoBK_Ca_ channel, quercetin, particulate matter

## Abstract

Particulate matter (PM) exposure increases reactive oxygen species (ROS) levels. It can lead to inflammatory responses and damage of the mitochondria thus inducing cell death. Recently, it has been shown that potassium channels (mitoK) located in the inner mitochondrial membrane are involved in cytoprotection, and one of the mechanisms involves ROS. To verify the cytoprotective role of mitoBK_Ca_, we performed a series of experiments using a patch-clamp, transepithelial electrical resistance assessment (TEER), mitochondrial respiration measurements, fluorescence methods for the ROS level and mitochondrial membrane potential assessment, and cell viability measurements. In the human bronchial epithelial cell model (16HBE14σ), PM < 4 μm in diameter (SRM-PM4.0) was used. We observed that PM decreased TEER of HBE cell monolayers. The effect was partially abolished by quercetin, a mitoBK_Ca_ opener. Consequently, quercetin decreased the mitochondrial membrane potential and increased mitochondrial respiration. The reduction of PM-induced ROS level occurs both on cellular and mitochondrial level. Additionally, quercetin restores HBE cell viability after PM administration. The incubation of cells with PM substantially reduced the mitochondrial function. Isorhamnetin had no effect on TEER, the mitoBK_Ca_ activity, respiratory rate, or mitochondrial membrane potential. Obtained results indicate that PM has an adverse effect on HBE cells at the cellular and mitochondrial level. Quercetin is able to limit the deleterious effect of PM on barrier function of airway epithelial cells. We show that the effect in HBE cells involves mitoBK_Ca_ channel-activation. However, quercetin’s mechanism of action is not exclusively determined by modulation of the channel activity.

## 1. Introduction

Many studies focus on intracellular potassium ion transport with particular attention to mitochondria [1,2]. It has been reported that changes in potassium flux through the inner mitochondrial membrane are essential for the induction of protective mechanisms diminishing tissue injuries caused by ischemia/reperfusion or oxidative stress [3,4]. Various observations suggest that the activation of the mitochondrial ATP-sensitive potassium (mitoK_ATP_) channel or mitochondrial large-conductance Ca^2+^-regulated potassium (mitoBK_Ca_) channel is a critical step in the induction of cardioprotective mechanisms [5,6,7]. Similarly, it has been reported that both channels participate in the neuroprotection of brain tissue against stroke-induced damage [8,9]. Additionally, it has been shown that activation of the mitochondrial potassium (mitoK) channels cause changes in the volume of the mitochondrial matrix, mitochondrial respiration, and membrane potential [10]. All these events lead to the participation of the mitoK channels in the regulation of mitochondrial reactive oxygen species (ROS) synthesis [11]. Therefore, it has been suggested that activation of both channels preserves mitochondrial function leading to increased cell survival.

The pharmacology of the BK_Ca_ channel family is well described. However, there is a continuous search for new selective activators which increase potassium influx into the mitochondrial matrix leading to cytoprotection. BK_Ca_ channel activators include synthetic substances such as NS1619 and NS11021, and natural substances such as quercetin and naringenin. To support the activator role of quercetin, an analog of quercetin—isorhamnetin, a substance which has one hydroxyl group changed to a methoxy group, is widely used [12]. Additionally, the known BK_Ca_ channel inhibitors that inhibit the activity of BK-type channels include paxilline, iberiotoxin, and prenitrem A [13,14,15,16].

The roles of natural substances in the regulation of ion channels are still under investigation. It is commonly known that they exhibit a large variety of biological properties. Quercetin acts as one of the strongest ROS scavengers among the flavonoids [17], it also reduces cellular damage caused by ROS [18]. What is more, quercetin regulates the expression of detoxification enzymes. It prevents LPS-induced myocardium damage by increasing SOD and catalase levels and reducing malondialdehyde levels [19].

The pharmacological modulation of mitochondrial potassium channels has become a promising approach treating cardiovascular or neurodegenerative diseases. However, the participation of the mitochondrial potassium channels in the cytoprotection of epithelial tissues, e.g., bronchial, is still not confirmed.

The respiratory epithelium is the primary site of particulate matter (PM) deposition. Thus, the mitoK channels of epithelial cells might be potential triggers of the cytoprotection mechanism against PM-induced injury. Unfortunately, currently our knowledge about the mitoK channels in bronchial tissue is limited since this topic has not been studied in detail. Our data suggest that epithelial cells contain functional mitoK channels [20]. However, their role in epithelial cytoprotection remains unknown, especially in the context of damage caused by PM or oxidative stress.

The source of particulate matter (PM), the main component of air pollution, can be both anthropogenic and natural. However, the anthropogenic source is dominant. PM exposure is associated with different health outcomes, such as premature death, asthma and its exacerbations, COPD, chronic bronchitis, cardiovascular disease, diabetes, and cancer [21]. PM’s adverse effects on human health are mainly linked to excess reactive oxygen species (ROS) formation and redox imbalance within the cell. The elevated ROS level has been linked to cytochrome P450 activity in the endoplasmic reticulum, NADPH-dependent P450 reductase activity in microsomes, electron leakage in mitochondria, and NADPH oxidase activity in phagosomes [22].

Since the respiratory epithelium protects the airway lumen against environmental agents such as allergens, pathogens, and contaminants by forming physical, mucosal, and innate immune barriers, epithelial integrity is fundamental [23]. Epithelial cell integrity is maintained by apical junctional complexes such as tight (TJs) and adherent junctions [24]. Numerous in vitro and in vivo studies show that particulate matter causes the disruption of airway epithelial integrity by downregulation of TJs, a decline in transepithelial electrical resistance (TEER), and an increase of paracellular permeability. These phenomena may contribute to the development of pulmonary diseases such as asthma, chronic obstructive pulmonary disease (COPD), chronic rhinosinusitis, and allergic rhinitis [25,26,27]. Thus, protecting the airways from pathogens and restoration of the epithelial barrier function might benefit airway disease treatment.

In this study, we reported the effect of quercetin on mitochondrial function in human bronchial epithelial cells exposed to particulate matter, with the implication of the mitochondrial large-conductance Ca^2+^-regulated potassium (mitoBK_Ca_) channels. To support the functional role of the mitoBK_Ca_ channel, we performed a series of experiments using the patch-clamp technique, transepithelial electrical resistance assessment, mitochondrial respiration measurements with the use of an oxygen electrode, fluorescent methods for the ROS level and mitochondrial membrane potential assessment, as well as cell viability measurements using trypan blue staining. We expect that these studies will bring us closer to a better understanding of the mitochondrial potassium channel activation-induced cytoprotective mechanisms in the respiratory epithelium.

## 2. Results

### 2.1. Quercetin Improves Transepithelial Resistance in HBE Cells

Transepithelial electrical resistance (TEER) was measured at 0, 30, 60, 90, and 180 min after exposure to PM. The diagram of substance administration at the apical side of cell monolayers during the measurements is presented in Figure 1a. As shown in Figure 1b, exposure to 50 µg/mL PM caused a decrease of TEER, 50 µM quercetin reversed the PM induced TEER decrease and penitrem A (300 nM) partially abolished this effect. Similarly, 3 µM NS11021, the mitoBK_Ca_ channel opener, also protected the cell monolayers from disruption, however, its effect was not as strong (Figure 1c). To confirm participation of the mitoBK_Ca_ channel in these phenomena, the analog of quercetin, isorhamnetin, was used. As was shown in Figure 1d, isorhamnetin did not exhibit a protective effect on the TEER of HBE cell monolayers exposed to PM. Taken together, these findings demonstrate that protective properties of quercetin against PM-induced cell damage could be linked with activation of the BK_Ca_-type channels. Additionally, results presented in Figure 1e do not indicate side effects of the substances used. The observed transepithelial electrical resistance changes in the presence of quercetin, isorhamnetin, NS11021, or penitrem A were not statistically significant.

### 2.2. Quercetin, but Not Isorhamnetin, Activates Mitochondrial BK_Ca_ Channel

The most straightforward method to assess the effect of quercetin on the mitoBK_Ca_ channel is a patch-clamp technique of mitoplasts at the single-channel level. Accordingly, a series of patch-clamp experiments were performed on the mitoplasts isolated from HBE cells. The effect of quercetin was measured at a concentration of 10 µM. All experiments were conducted in the presence of 100 µM calcium ions. It was shown that the open probability of the mitoBK_Ca_ channel increased after the admission of 10 µM quercetin. The result is illustrated by representative fragments of recordings obtained during patch-clamp measurements (Figure 2a). To confirm the channel’s identity, 300 nM penitrem A was added. After the admission of penitrem A, the mitoBK_Ca_ channel activity decreased by 50% and remained at a fully closed state 4 min after adding penitrem A (Figure 2a). As the control to this experiment, quercetin’s analog, isorhamnetin, was used to conduct patch-clamp experiments under the same conditions. Figure 2b shows that 10 µM isorhamnetin did not increase the channel’s activity. The open probability of the channels is presented in Figure 2c. We observed an increase in the activity of the mitoBK_Ca_ as an effect of 10 µM quercetin and a decrease after 300 nM penitrem A administration.

### 2.3. Quercetin Impacts Mitochondrial Respiration Rate in BK_Ca_ Channel Dependent Way

Due to the observed capability of quercetin to activate the mitoBK_Ca_ channel, its effect of quercetin on mitochondrial respiration rate was examined. To determine the respiration rate of HBE cells, high resolution respirometry was applied to measure oxygen consumption. The measurement was conducted in live non-permeabilized cells to analyze the effect of quercetin on mitoBK_Ca_ channel activation, reflected by changes in the respiration rate of HBE cells. It is believed that mitochondrial potassium channels affect the activity of the electron transport chain via increased flux of potassium ions across the inner mitochondrial membrane and dissipation of mitochondrial membrane potential. As shown, 1, 3, and 10 μM quercetin (Figure 3a) increased respiration rate of HBE cells in a dose-dependent manner (15.2% for 1 to 32.8% for 10 μM quercetin compared to control). This effect was not observed after administration of isorhamnetin (1, 3, 10, 30 μM) (Figure 3b). Additionally, as shown in the Figure 3a, the administration of 300 nM penitrem A before quercetin abolished the effect of quercetin, which suggests that the increase of HBE cells’ respiration rate by quercetin is associated with activation of the mitoBK_Ca_ channel. For the maximal respiration rate, the FCCP was administered. The maximal respiration was higher for quercetin-treated cells without a BK_Ca_ channel inhibitor, also implying the channel’s involvement in the respiration rate. However, the effect was not statistically relevant.

### 2.4. The Mitochondrial Potential of HBE Cells Is Affected by Quercetin

Previous studies have shown that mitochondrial potassium channel openers lead to mitochondrial membrane depolarization by the K^+^ influx to mitochondrial matrix [13,28]. Here, the effect of quercetin on mitochondrial membrane potential was examined with the use of the rhodamine 123 fluorescent dye. The experiment was conducted as described in the methods section and compounds were added to the chambers in the order presented in Figure 4a. Figure 4b indicates that quercetin (1, 3, 10, 30 μM) decreased mitochondrial membrane potential in a dose-dependent manner leading to depolarization. The most visible effect was seen at the highest concentration used, which caused a mitochondrial potential decrease of 19.6%. Isorhamnetin did not affect the mitochondrial potential (Figure 4c). For maximal uncoupling FCCP was applied.

### 2.5. Quercetin Demonstrates Antioxidant Properties on Cellular Level

It is well known that changes in mitochondrial activity lead to alterations in ROS levels. The antioxidant properties of quercetin and isorhamnetin were determined by H2DCFDA fluorescent staining. In this experiment, 50 μg/mL PM exposure’s effect on ROS levels in HBE cells was determined. Quercetin (1, 3, 10, 30 μM) (Figure 5a), as well as isorhamnetin (1, 3, 10, 30 μM) (Figure 5b) have shown antioxidant effects in HBE cells. The most potent antioxidant was quercetin at 10 μM concentration, which reduced ROS levels by 52%. The antioxidant effect in PM-treated cells was observed both for quercetin (Figure 5c) and isorhamnetin (Figure 5d). The antioxidant effect of quercetin is the most visible at 10 μM concentration, as the ROS level is lower by 44% than the level in PM-treated cells. Interestingly, 300 nM penitrem A did not abolish the effect of quercetin, which could lead to a conclusion that the antioxidant properties of quercetin on HBE cells was not related to mitoBK_Ca_ channel opening. This conclusion is favorable with the result that isorhamnetin (which did not activate the mitoBK_Ca_ channel) could also diminish the oxidative stress in HBE cells, both the control ones and those incubated with 50 μg/mL PM.

### 2.6. Quercetin Demonstrates Antioxidant Properties on Mitochondrial Level

The antioxidant effect of quercetin and its analog, isorhamnetin, was determined on mitochondrial level by MitoSOX fluorescent staining. The ROS level was measured both in control HBE cells and HBE cells after 50 μg/mL PM exposure. Figure 6 shows that quercetin (1, 3, 10, 30 μM) did not have any antioxidant effect on mitochondrial level in HBE cells, neither did isorhamnetin (1, 3, 10, 30 μM) (Figure 6b). 50 μg/mL PM increased mitochondrial ROS levels (Figure 6c,d). Quercetin and isorhamnetin had antioxidant properties on mitochondria of HBE cells in 50 μg/mL PM-induced oxidative damage. The antioxidant properties of quercetin were stronger than isorhamnetin’s. Similarly to the results on a cellular level, 300 nM penitrem A did not abolish the effect of quercetin, which indicates that mitoBK_Ca_ channels do not determine antioxidant properties of quercetin on mitochondria.

### 2.7. Long Term Application of PM Impairs Mitochondrial Function and Cell Viability

To assess cell viability of HBE cells after PM (50 μg/mL), quercetin (1, 3, 10, 30 μM), and isorhamnetin (1, 3, 10, 30 μM) exposure, trypan blue staining was performed. Cells were incubated with appropriate compounds for 24 h before the measurement. As shown on Figure 7a, 50 μg/mL PM exposure diminished cells viability. Quercetin partially abolished this effect and the restoration of cells viability was the most visible after incubation with 10 μM quercetin. The analog of quercetin, isorhamnetin, did not reverse the damage caused by 50 μg/mL PM.

Mitochondrial dysfunction was assessed by high resolution respirometry with subsequent determination of non-phosphorylating leaks (with inhibitor of ATPase, oligomycin, 4 μg/mL), maximum uncoupled electron transfer system (ETS) (with the uncoupler FCCP; 1 μM), and residual oxygen consumption (ROX) (with the inhibitor of ETS complexes I and III, rotenone, 2.5 μM and antimycin A, 0.5 μM). Figure 7b shows that baseline respiration of HBE cells is significantly lower (by approximately 62%) in cells incubated with 50 μg/mL PM for 24 h. ETS was also reduced (by 61%) after PM exposure, which suggests that mitochondrial functions are defective after PM administration. Interestingly, co-exposure with PM and quercetin (1, 3 μM) did not restore mitochondrial function (Figure 7b).

## 3. Discussion

Flavonoids are under investigation since their protective effects against cell damage of different origins are well known. One of the most investigated flavonoids is quercetin, which exhibits many protective effects, e.g., against ischemia-reperfusion injury [29], neurogenerative diseases [30,31], cancer [32], or in bone homeostasis [33]. Recent studies indicate that quercetin is potent in protecting against airborne PM-induced toxicity in various models [34,35]. As expected, in our experimental model, PM disrupted the epithelial integrity as shown by TEER measurements, which is consistent with the findings of other groups working on endothelial [36] and epithelial cells [37]. Our further studies showed that the effect might be associated with mitoBK_Ca_ and could be regulated by quercetin, but not by its analog isorhamnetin. The latter served as a control flavonoid to quercetin in our further studies. The possible relationship between quercetin, PM, and mitoBK_Ca_ was further investigated. To support our data, NS11021, a classical BK_Ca_ channel opener was used.

Mitochondrial potassium channels are well known for their protective effects, and it is well documented that mitoBK_Ca_ activation is involved in protection against cardiac injury [38]. For the first time, we show that the mitoBK_Ca_ channel is present in the inner mitochondrial membrane of the human bronchial epithelial cell line, and the channel’s activity is modulated by quercetin. Regulation of mitoBK_Ca_ channel by flavonoids was also confirmed in the mitochondria of endothelial cells [16]. In our results, it was additionally observed that isorhamnetin failed to activate mitoBK_Ca_. The indicated capability of quercetin to increase the open probability of the channel led to the investigation of mitochondrial activity.

The primary mitochondrial function is the energy production with simultaneous oxygen consumption. Thus, the effect of quercetin was examined in terms of respiration rate. So far, the results obtained from different laboratories are ambiguous. It was described by Dorta et al. [39] that quercetin increases state 3 respiration but has no effect on state 4 respiration of isolated mitochondria from rat liver. We incorporated the whole-cell model and observed an increase in the respiration rate of HBE cells after quercetin administration. We assume that the increase in oxygen consumption rate may be linked to mitoBK_Ca_ activation observed in patch-clamp measurements since penitrem A reversed the effect. The mechanism underlying this phenomenon is explained by increased potassium influx to the mitochondrial matrix after channel activation resulting in partial dissipation of mitochondrial potential and increased electron transport chain activity. We confirmed this mode of action of quercetin by measuring mitochondrial potential with the fluorescent probe rhodamine 123 (which was not interacting with compounds used in the experiments) and showing that quercetin decreased mitochondrial membrane potential in a dose-dependent manner. Previous studies using another fluorescent probe (JC-1) indicated a slight drop in mitochondrial membrane potential upon 8 μM quercetin administration [34]. Our control analog of quercetin, isorhamnetin, failed to change the oxygen consumption rate and the mitochondrial membrane potential. The observation disagrees with the platelet experiments’ results, which show that isorhamnetin decreases mitochondrial membrane potential [40]. However, the discrepancies may result from different methods and cell models implemented in the studies. The platelet study involved the TMRE method on whole-cell suspension. In contrast, in our study, the environment for the experiment was strictly controlled (using inhibitors of complex I and ATP-synthase and succinate as the substrate), thus eliminating the activity of some proteins that could affect the results. Other data from the Jin group [34] indicated the possible role of quercetin in reversing the PM-induced drop in mitochondrial membrane potential, which led us to investigate the mechanisms that may be involved in the protective effect and correlated to mitochondrial activity, namely ROS production.

It is well known that PM-induced toxicity is strongly related to ROS overproduction leading to an inflammatory response and development and/or exacerbation of diseases such as asthma, cancers, lung infections, and cardiovascular diseases [41,42,43]. Recent reports demonstrated that quercetin diminishes the PM-induced ROS production measured as total ROS generated within the cell [34]. Additionally, we showed that both quercetin and isorhamnetin caused a decrease in ROS levels, both on basal and PM-induced levels. However, this data reflects the total ROS level within the cell. Considering the fact that quercetin activated mitoBK_Ca_, it is crucial to explore the quercetin effects on mitochondrial-derived ROS. It was previously reported that quercetin has an antioxidant effect on membrane lipid peroxidation in isolated mitochondria [44]. Our results show that quercetin at higher concentrations is potent enough to increase the basal mitochondrial ROS level in untreated HBE cells. The observed effect may be related to harmful activities of quercetin and its metabolites at high concentrations [45]. Some studies indicate that quercetin may interact with mPTP (mitochondrial permeability transition pore) and act both as an mPTP inhibitor (at low concentrations, ~5 μM) or an mPTP opener (at high concentrations > 40 μM) [46]. The observed phenomenon may explain the increase in mitochondrial ROS level after administration of high concentrations of quercetin to untreated cells. In turn, in PM-treated cells, quercetin was more potent in reducing ROS than isorhamnetin. However, the effect was not reversed by penitrem A, suggesting other targets of quercetin’s action that are not associated with mitoBK_Ca_.

The observed antioxidant activity of quercetin in PM-treated cells and its ability to activate mitoBK_Ca_ may explain our observations that quercetin restores the cell viability of HBE cells treated with PM. Interestingly, PM strongly reduced mitochondrial capacity measured by the oxygen consumption rate. According to studies of other groups, quercetin, but not isorhamnetin, increases mitochondrial respiration [47]. In our studies, neither quercetin nor isorhamnetin was able to restore mitochondrial function after PM administration, thus their role in this matter seems plausible.

Our study has limitations. Bronchial epithelium is composed of variety of cell types, each playing different functions. For our experiments, the widely utilized for a variety of epithelial physiological and disease processes 16HBE14o- cell model was chosen as it retains many of the functions and morphology of differentiated normal bronchial epithelium. However, it must be noted that the immortalized cell lines may have different characteristics from primary cells and the effects of quercetin on different cell types present in airway epithelium, including stem/progenitor cells [48] remain to be investigated.

In summary, we demonstrated that one of the quercetin targets is mitoBK_Ca_, and some of the mitochondrial functions are related to this phenomenon. However, it is noteworthy that there are many quercetin targets within the cell. It was demonstrated that quercetin might inhibit or open the mPTP, depending on the concentration [46]. It may also be involved in inhibiting ATP-ase but not the ATP-synthase activity of complex V in mitochondria [49]. Additionally, it was reported that quercetin might inhibit complex I in mitochondria [50] or restore its function [51]. In further studies, it is essential to acknowledge all known targets in the quercetin mode of action, considering the possible effects associated with mitoBK_Ca_ opening.

## 4. Materials and Methods

### 4.1. Cell Culture

The experiments were performed on human bronchial epithelial cell line (16HBE14σ) purchased from MERCK (Darmstadt, Germany). The HBE cells were cultured in Minimal Essential Medium Eagle (MEM; MERCK (Darmstadt, Germany)) supplemented with 10% fetal bovine serum (FBS; Gibco (Thermo Fisher Scientific, Waltham, MA, USA)) and antibiotics (100 U/mL penicillin, 100 mg/mL streptomycin). Cells were grown in the cell incubator at 37 °C in a humidified atmosphere with 5% CO_2_. Cells were passaged twice a week after reaching 90% confluence.

### 4.2. Chemicals

Quercetin, penitrem A, and isorhamnetin were purchased from MERCK (Darmstadt, Germany). Particulate matter PM < 4 μm SRM-2786 (PM) was obtained from Nation Institute of Standards and Technology (NIST, Gaithersburg, MD, USA). In our study, we used PM according to NIST standard in order to ensure the repeatability of biophysical and biochemical experiments. PM was formulated from atmospheric particulate material gathered in 2005 from an air intake filtration system of a major exhibition center in Prague, Czech Republic. As the producer declares, the PM sample is not expected to represent the area from which it was collected, it should rather reflect atmospheric particulate matter present in an urban area.

### 4.3. Transepithelial Electrical Resistance (TEER) Measurements

For barrier integrity measurements the cells were seeded onto Corning Costar Snapwell inserts (0.45 μm, 1.12 cm^2^ surface area) and grown as described previously [52]. Briefly, the cells were grown under liquid–liquid interface (LLI) followed by the air–liquid interface (ALI) to polarize the cells. Prior to each experiment, the cell culture medium was replaced by sterile Ringer solution both on the apical (0.5 mL) and basolateral (3.5 mL) side of the cell monolayer followed by 1 h of incubation at 37 °C and 5% CO_2_ atmosphere. Next the cell layer resistance was measured using EVOM2 voltohmmeter with STX-2 chopstick electrodes (WPI). To calculate the monolayer resistance, the resistance of blank insert (cell-free Snapwell) was subtracted from each TEER reading. The cell layers that did not reach a resistance higher than 300 Ω cm^2^ were not considered for further experiments. Cell monolayers were then pretreated with quercetin, penitrem/quercetin, isorhamnetin, penitrem/isorhamnetin, NS11021, penitrem/NS11021, or the carriers (control) for 1 h followed by the addition of PM to apical compartment (at final concentration of 50 μg/mL) and measurement of TEER (T = 0). TEER of monolayers was then measured after 30, 60, 90, and 180 min after PM introduction.

### 4.4. Mitoplast Preparation for Electrophysiology

Mitochondria and mitoplast for electrophysiological measurements (patch-clamp method) were prepared as previously described [53,54]. Shortly, cells underwent differential centrifugation and subsequently hypotonic swelling. First, cells were harvested after reaching 90% confluence, suspended in 3.5 mL PBS (Lonza) per bottle, and centrifuged at 400× *g* for 5 min. After centrifugation, supernatant was discarded and pellet was resuspended in 2 mL preparation solution (250 mM sucrose, 5 mM HEPES, pH 7.2). The cells were homogenized (glass-glass homogenizer no 19, Kontes Glass) and centrifuged at 9200× *g* for 10 min in 4 °C. The pellet was resuspended in preparation solution and centrifuged for 10 min in 750× *g* to separate the fraction of purified mitochondria. Supernatant containing mitochondria was centrifuged for 10 min at 9200× *g*. The pellet with isolated mitochondria was suspended in storage solution (150 mM KCl, 10 mM HEPES, pH 7.2). In order to prepare the mitoplasts, isolated mitochondria of HBE cells were incubated for 1 min in a hypotonic solution (5 mM HEPES, 100 μM CaCl_2_, pH 7.2). Afterwards, the hypertonic solution (750 mM KCl, 30 mM HEPES, and 100 μM CaCl_2_, pH 7.2) was added for restoration of the isotonicity of the medium.

### 4.5. Patch-Clamp Experiments

Fresh mitoplasts were prepared for each patch-clamp experiment. Patch-clamp experiments on HBE mitoplasts were performed using patch-clamp pipette filled with an isotonic solution (150 mM KCl, 10 mM HEPES, and 100 μM CaCl_2_ at pH 7.2). Substances, the effect of which we examined on mitoBK_Ca_ channels (quercetin, penitrem A, isoramnetin, PM), were diluted in isotonic solution and applied into the measurement by a perfusion system. For patch-clamp experiments we used pipettes made of borosilicate glass, which had resistance of 10–20 MΩ (Harvard Apparatus GC150-10, Holliston, MA, USA). The current–time traces of the experiments were recorded in a single-channel mode. The current–voltage relationship was used to establish the conductance of examined channel. We determined open probability of the channel (NPo) using the single-channel search mode of the Clampfit 10.7 software (Axon Instruments, Molecular Device, San Jose, CA, USA).

### 4.6. High-Resolution Respirometry

High resolution respirometry was performed using Oxygraph-2K system (Oroboros Instruments, Innsbruck, Austria) according to an established method [55]. Shortly, HBE cells were harvested, centrifuged, and resuspended in MEM in 1 × 10^6^ cells/cm^3^. Cells were added to respiratory chambers to measure routine respiration. Quercetin (1, 3, or 10 μM), isorhamnetin (1, 3 or 10 μM), or penitrem A (300 nM) were added to chambers after stabilization of the respiration. The maximal uncoupled electron transfer system (ETS) was measured using carbonyl cyanide-p-trifluoromethoxyphenylhydrazone (FCCP; 1 μM).

### 4.7. Potential Measurements

Potential of the inner mitochondrial membrane was determined using O2k-FluoRespirometer fluorometrically (Oroboros Instruments, Innsbruck, Austria) with the use of the fluorophore rhodamine 123 (1.3 µM). HBE cells were harvested, centrifuged, and resuspended in mitochondrial respiration medium, MiR05 (0.5 mM EGTA, 3 mM MgCl_2_, 60 mM lactobionic acid, 20 mM taurine, 10 mM KH_2_PO_4_, 20 mM HEPES, 110 mM D-sucrose, 1 g/L BSA, fatty acid free) at 1 × 10^6^ cells/cm^3^ concentration. Cells were added to the respiratory chambers to measure routine respiration along with mitochondrial membrane potential. Digitonin (10 µg/mL) was added in order to permeabilize cells, rotenone (0.5 µM) was added to block the complex I in mitochondria. Succinate (2 mM) was added as respiratory substrate. Oligomycin (4 mg/mL) was added after adding ADP (2.5 mM) to block ATP synthase. Afterwards, quercetin (1, 3, 10, or 30 μM) or isorhamnetin (1, 3, 10, or 30 μM) were added to examine their effect on mitochondrial membrane potential. To measure maximal uncoupling, FCCP 1 µM was added.

### 4.8. Measurement of Reactive Oxygen Species

Cells were harvested after reaching 90% confluence and resuspended in culture medium at concentration 500,000 cells/mL. A suspension of cells was added to a 96-well plate (50,000 cells/well). Cells were grown for 24 h in a cell incubator in the same conditions. The following day, medium was removed, cells were washed with PBS and incubated for 30 min with 25 µM 2′,7′-Dichlorofluorescin diacetate (H2DCFDA; Invitrogen (Thermo Fisher Scientific, Waltham, MA, USA)) for total ROS measurements or with MitoSOX Red (Molecular Probes, (Thermo Fisher Scientific, Waltham, MA, USA)) for mitochondrial ROS measurement. After the incubation, 50 mg/mL PM was added along with different concentrations of quercetin, isorhamnetin, and penitrem A. Cells were incubated for 3 h and fluorescence was measured using Fluoroskan Ascent (Thermo Fisher Scientific, Waltham, MA, USA) at ex/em wavelengths: 485/538 nm (H2DCFDA) and 485/590 nm (MitoSOX Red).

### 4.9. Trypan Blue Staining for Cell Viability Assessment

The cytotoxicity of PM was investigated using Trypan Blue staining. HBE cells were incubated with PM along with quercetin, isorhamnetin, or penitrem A at various concentrations for 24 h. After the incubation, cells were detached with trypsin, then resuspended in medium and incubated with 1:1 Trypan Blue dye for 3 min. Cell survival was measured in a TC20 cell counter (BioRad, Hercules, CA, USA).

### 4.10. Mitochondrial Dysfunction Assessment

In order to measure mitochondrial dysfunction, high resolution respirometry was applied using Oxygraph-2K system (Oroboros Instruments, Innsbruck, Austria). Before the measurement, the cells were incubated with 50 µg/mL PM, 50 µg/mL PM, and 1 µM quercetin, or with 50 µg/mL PM and 3 µM quercetin for 24 h prior the experiment. After harvesting, HBE cells were centrifuged and resuspended in MEM at 1 × 10^6^ cells/cm^3^ concentration. Then the cells were added to the respiratory chambers to measure routine respiration. The procedure was performed according to previously described method [56]. Briefly, HBE cells were exposed to 4 µg/mL oligomycin, 1 µM FCCP, and 1 µM rotenone with 5 µM antimycin A to measure mitochondrial dysfunction in HBE cells. Non-phosphorylating leaks were determined with oligomycin (4 μg/mL). Maximum uncoupled electron transfer system (ETS) was measured using carbonyl cyanide-p-trifluoromethoxyphenylhydrazone (FCCP; 1 μM), residual oxygen consumption (ROX) was determined by addition of rotenone (2.5 μM) and antimycin A (0.5 μM).

## 5. Conclusions

A better understanding of the relationships between cell physiology, mitochondrial metabolism, and regulation of the mitoBK_Ca_ channel can help in the development of effective strategies for cytoprotection. Probably, by being on the trail of one of the oldest cell protection mechanisms, we can learn how to improve the treatment of damage induced by particulate matter.

## Figures and Tables

**Figure 1 ijms-24-00638-f001:**
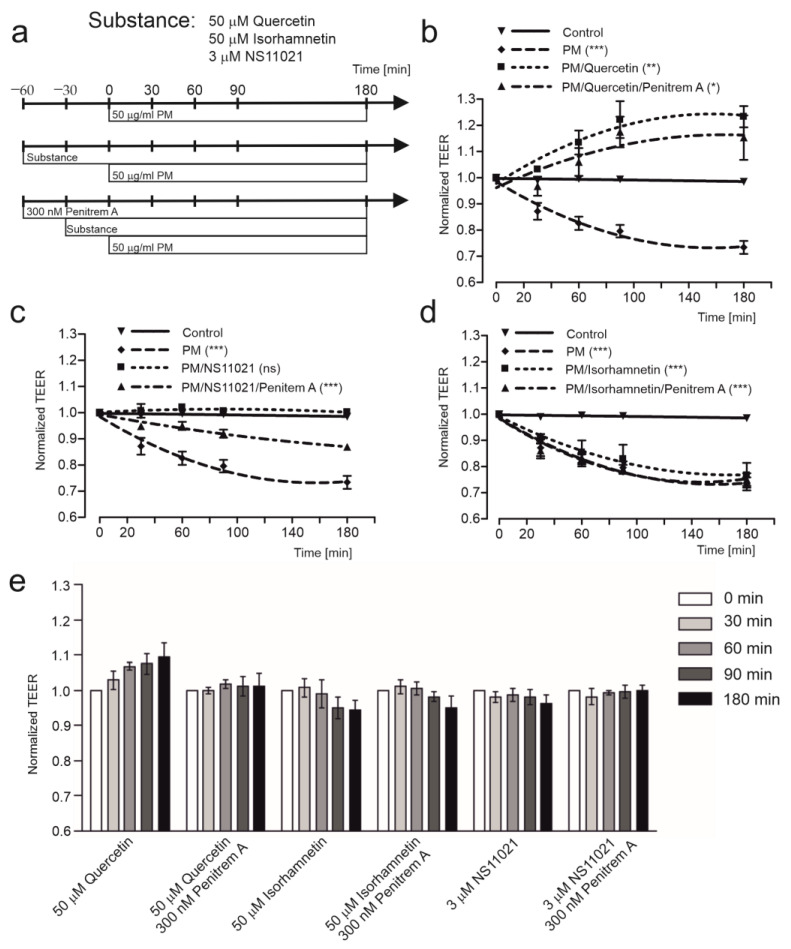
Transepithelial electrical resistance changes of 16HBE14o-cell monolayers. (**a**) Schematic diagram of the experimental protocol. (**b**) TEER change following exposure to Ringer solution alone (control), 50 µg/mL PM with/without 50 µM quercetin, and 300 nM penitrem A. (**c**) TEER change following exposure to Ringer solution alone (control), 50 µg/mL PM with/without 3 nM NS11021, and 300 nM penitrem A. (**d**) TEER change following exposure to Ringer solution alone (control), 50 µg/mL PM with/without 50 µM isorhamnetin, and 300 nM penitrem A. (**e**) TEER change following exposure to Ringer solution supplemented with 50 µM Quercetin, 50 µM Quercetin, and 300 nM penitrem A, 50 µM Isorhamnetin, 50 µM Isorhamnetin, and 300 nM penitrem A, and 3 nM NS11021, 3 nM NS11021, and 300 nM penitrem A. Data were presented as the mean ± SD of experiments conducted independently (*n* = 4). Statistical significance of measurements compared to controls was determined at *p* < 0.05 (*), *p* < 0.01 (**), and *p* < 0.001 (***) by one-way ANOVA.

**Figure 2 ijms-24-00638-f002:**
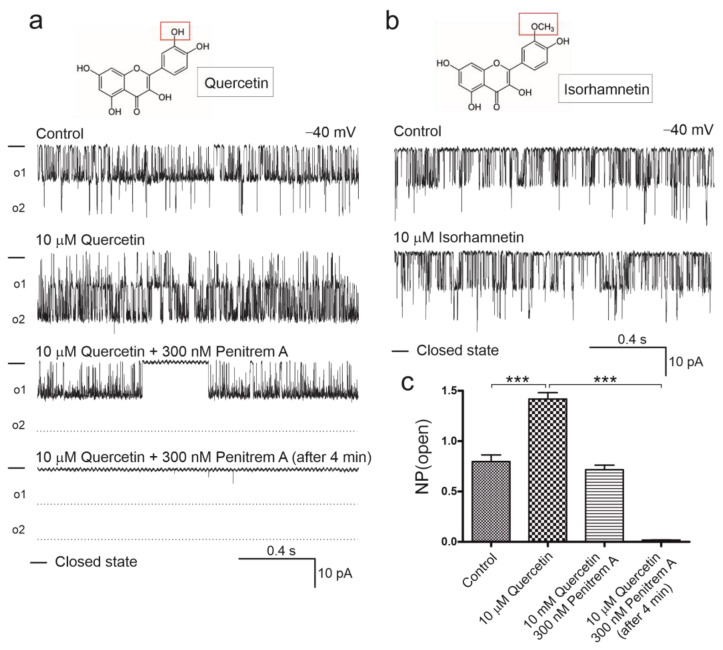
Assessment of mitoBK_Ca_ channel activity after application of quercetin, isorhamnetin, and penitrem A. (**a**) Chemical structure of quercetin and single channel recordings of activating effect of 10 µM quercetin. An amount of 300 nM penitrem A inhibits the mitoBK_Ca_ channel activity after activation by quercetin. Full inhibitory effect of penitrem A is observed after 4 min. (**b**) Chemical structure of isorhamnetin and representative recordings indicate lack of 10 µM isorhamnetin effect on mitoBK_Ca_ channel activity. (**c**) Analysis of the mitoBK_Ca_ channel open probability after application of quercetin at applied potential of −40 mV (mean ± SD, *n* = 3). Statistical significance was determined at *p* < 0.001 (***) by one-way ANOVA.

**Figure 3 ijms-24-00638-f003:**
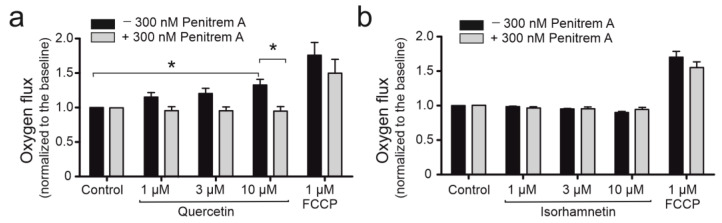
The effect of quercetin and isorhamnetin on oxygen consumption rate in whole HBE cell model. The bars represent the effect of quercetin at different concentrations (1, 3, 10 µM) (**a**), and isorhamnetin (1, 3, 10 µM) (**b**), on HBE cell oxygen consumption rate, measured with high-resolution respirometry. The effect of the tested compounds (black bars) was compared to oxygen consumption rate measured in the presence of 300 nM penitrem A (gray bars). Full uncoupling was induced by 1 µM FCCP. The results are represented by means ± SD, *n* = 3. Statistical significance was determined at *p* < 0.05 (*) by one-way ANOVA.

**Figure 4 ijms-24-00638-f004:**
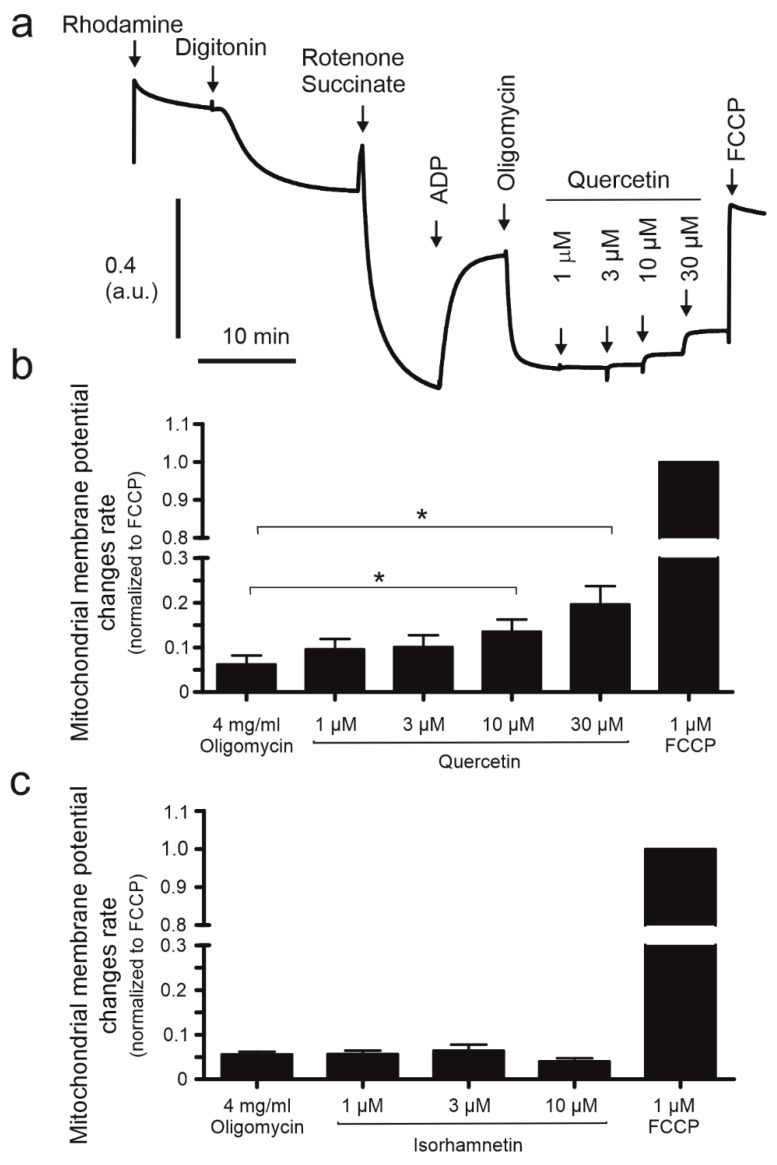
The effect of quercetin and isorhamnetin on mitochondrial membrane potential. (**a**) Representative recording of mitochondrial membrane potential after application of 1.3 µM rhodamine, 10 µM digitonin, 0.5 µM rotenone, 2 mM succinate, 2.5 µM ADP, 4 µg/mL oligomycin, quercetin (1, 3, 10, 30 µM), and finally, 1 µM FCCP. Below, the bars represent the effect of quercetin at different concentrations (1, 3, 10, 30 µM) (**b**) and isorhamnetin (1, 3, 10 µM) (**c**) on mitochondrial membrane potential. The results are represented by means ± SD, *n* = 3. Statistical significance was determined at *p* < 0.05 (*) by one-way ANOVA.

**Figure 5 ijms-24-00638-f005:**
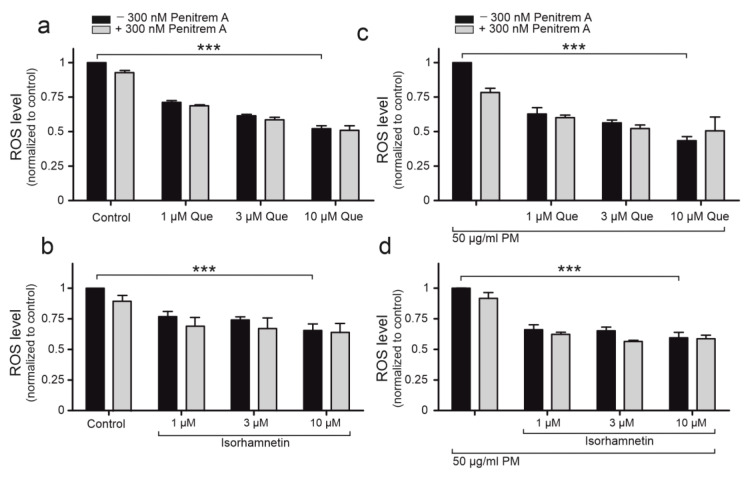
The influence of quercetin and isorhamnetin on intracellular ROS level. Black bars represent the effect of quercetin (1, 3, 10 µM) (**a**) and isorhamnetin (1, 3, 10 µM) (**b**) on intracellular ROS level. Influence of 50 µg/mL PM on ROS intracellular level along with quercetin (1, 3, 10 µM) (**c**), isorhamnetin (1, 3, 10 µM) with 50 µg/mL PM (**d**). Gray bars represent the data after application of 300 nM penitrem A. Data is normalized to the control values and expressed as means ± SD, *n* = 3. Statistical significance was determined at *p* < 0.001 (***) by one-way ANOVA.

**Figure 6 ijms-24-00638-f006:**
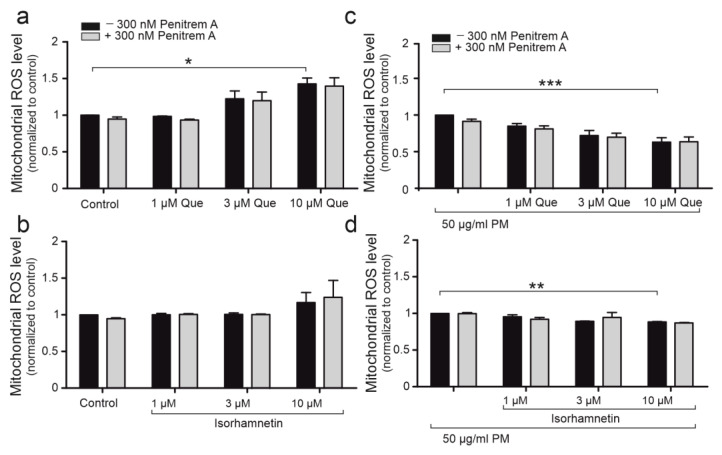
The influence of quercetin and isorhamnetin on mitochondrial ROS levels. It has been presented effect of quercetin (1, 3, 10 µM) (**a**) and influence of isorhamnetin (1, 3, 10 µM) (**b**) on mitochondrial ROS level measured using MitoSOX fluorescent dye. The effect of application of 50 µg/mL PM witch quercetin (1, 3, 10 µM) (**c**) and isorhamnetin (1, 3, 10 µM) (**d**) on mitochondrial ROS level were reported. The influence of 300 nM Penitrem A is represented by gray bars. Data is normalized to the control values. All results are represented by means ± SD, *n* = 3. Statistical significance was determined at *p* < 0.05 (*), *p* < 0.01 (**), *p* < 0.001 (***) by one-way ANOVA.

**Figure 7 ijms-24-00638-f007:**
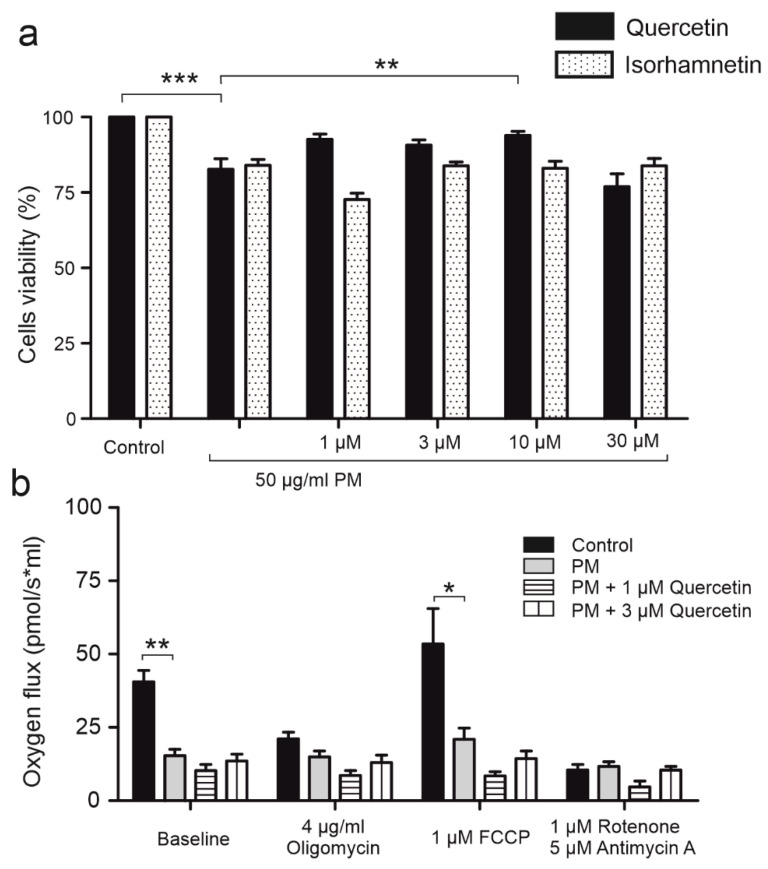
Effect of PM, quercetin, and isorhamnetin on HBE cell’s viability and mitochondrial function. (**a**) The bars represent the effect of 50 µg/mL PM and quercetin (1, 3, 10, and 30 µM) on cell viability measured using Trypan blue staining. Cell viability was measured after 24 h incubation with testing compounds. The effect is compared to control viability of the untreated cells. (**b**) HBE cells were exposed to subsequently 4 µg/mL oligomycin, 1 µM FCCP, and 1 µM rotenone with 5 µM antimycin A to measure mitochondrial dysfunction in control cells (black bars), cells incubated 24 h with 50 µg/mL PM (gray bars), cells incubated 24 h with 50 µg/mL PM and 1 µM quercetin (horizontal line bars), cells incubated 24 h with 50 µg/mL PM and 3 µM quercetin (vertical line bars). The results are represented by means ± SD, *n* = 3. Statistical significance was determined at *p* < 0.05 (*), *p* < 0.01 (**), *p* < 0.001 (***) by one-way ANOVA.

## Data Availability

Not applicable.

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
