# Peer review of "Effect of Quercetin on mitoBKCa Channel and Mitochondrial Function in Human Bronchial Epithelial Cells Exposed to Particulate Matter"

_ijms, 2022, doi:10.3390/ijms24010638_

Round 1
Reviewer 1 Report
The study A. Dabrowska et al evaluated the implication of potassium channel in the mitochondria (mitoK) in the response to particulate matter (PM) and the protective effect of Quercetin. This is an important question due to the deleterious effect of air pollution on airways. The study is original and the data are convincing. I have some comments and suggestions in order to improve the article.
Major comments
1. How was determined the dose of PM used for HBE exposure? Is it related to the level of PM present in polluted air according to the legal alarm level. I am not sure that the PM 2786 reflect the urban airway pollution present in our environment. This should be discussed at least.
2. Line 104: I think that the data did not support this conclusion. Indeed, the protective effect of quercetin is not modulated by addition of PenitremA. It is unlikely that the effect of quercetin is mainly related to mitoK activation.
3. In the figure 3, the authors did not evaluate the oxygen consumption after exposure to PM. It seems important to evaluate this parameter in this condition of activation.
4. In the figures 5 and 6, the authors reported the intracellular and mitochondrial ROS levels at baseline and after PM exposure. The data are reported as normalized value compared to the control. In the figure 5A and B, the value are about 4 and 10 whereas all the other panels, the control are equal to 1. The authors should report all the data with the same calcul (preferentially, as the absolute level of ROS). This will allow to evaluate the effect of PM on this parameter.
5. It seems important to mention in the introduction that quercetin has been showed to upregulate the expression of detoxification enzymes and is able to act as a scavenger or reactive oxygen species. This should be also discussed.
Minor comments
1. line 24: Please reworded this sentence. “Quercetin is able to limit the deleterious effect of PM on barrier function of airway epithelial cell” or something like this.
Author Response
File author-coverletter-24901343.v1.docx contain our responces to you comments.

Reviewer 2 Report
Dabrowska, et al. investigated the protective role of quercetin, a mitoBKCa channel opener, in human bronchial epithelial cells exposed to particulate matter by using a couple of tools including patch-clamp technique, transepithelial electrical resistance assessment, mitochondrial respiration measurements, fluorescent methods for the ROS level and mitochondrial membrane potential assessment, as well as cell viability measurements using trypan blue staining. The manuscript was well written. The data are novel and interesting. There are some concerns.
1. The authors should provide components of particulate matter and discuss what components have the obtained effects.
2. As an analogue of quercetin, isorhamnetin seemed to act differently in some experiments. Therefore, isorhamnetin must be introduced in the section of Introduction in terms of similarities and differences between quercetin and isorhamnetin. Provide the rational why isorhamnetin was used.
3. Line 100, Fig.1d should be Fig.1c? It seemed that authors talked about NS11021.
4. It seemed that Fig. 1e was not described in the manuscript.
5. In Fig2c, is there any significance between treatment and control?
6. Authors could consider not to label “ns” if there is no significance.
7. Authors should mention limitations of the study. The human bronchial epithelial cell line is usually immortal and different from primary cells. In addition, there are number of epithelial cells in the airway (PMID:35107300), therefore authors should discuss these cells including stem progenitor cells in terms of effects of quercetin.
Author Response
File author-coverletter-25038912.v1.docx contains our responces.

Round 2
Reviewer 1 Report
As mentionned in my first reviewing, I do not understand the mode of expression of the data in figure 5 and 6 (comment 4). In the revised version, I cannot clearly see the changes performed by the authors. The data are claimed to be reported as normalized value compared with the control although in some figures, the value of the control are about 4 or 10. Please clearly explained the mode of calcul for each part of these figures or showed these data as the absolute level of oxygen metabolites.
Author Response
Attached file contain our responces to you comments.
